# Construction of Structured Random Measurement Matrices in Semi-Tensor Product Compressed Sensing Based on Combinatorial Designs

**DOI:** 10.3390/s22218260

**Published:** 2022-10-28

**Authors:** Junying Liang, Haipeng Peng, Lixiang Li, Fenghua Tong

**Affiliations:** 1Information Security Center, State Key Laboratory of Networking and Switching Technology, Beijing University of Posts and Telecommunications, Beijing 100876, China; 2National Engineering Laboratory for Disaster Backup and Recovery, Beijing University of Posts and Telecommunications, Beijing 100876, China; 3Shandong Provincial Key Laboratory of Computer Networks, Shandong Computer Science Center (National Supercomputer Center in Jinan), Qilu University of Technology (Shandong Academy of Sciences), Jinan 250014, China

**Keywords:** compressed sensing, semi-tensor product, measurement matrices, incidence matrices, embedding operation, coherence

## Abstract

A random matrix needs large storage space and is difficult to be implemented in hardware, and a deterministic matrix has large reconstruction error. Aiming at these shortcomings, the objective of this paper is to find an effective method to balance these performances. Combining the advantages of the incidence matrix of combinatorial designs and a random matrix, this paper constructs a structured random matrix by the embedding operation of two seed matrices in which one is the incidence matrix of combinatorial designs, and the other is obtained by Gram–Schmidt orthonormalization of the random matrix. Meanwhile, we provide a new model that applies the structured random matrices to semi-tensor product compressed sensing. Finally, compared with the reconstruction effect of several famous matrices, our matrices are more suitable for the reconstruction of one-dimensional signals and two-dimensional images by experimental methods.

## 1. Introduction

In the era of data explosion, with the increasing amount of information, data acquisition, transmission and storage devices are facing increasingly severe pressure. At the same time, the data processing process will also be accompanied by the risk of information disclosure. The loss of some data may threaten the safety of life and property, and now, data disclosure is common. Therefore, in the era of big data, people urgently need to find a new data processing technique to decrease the risk of data leakage during information processing and release the pressure of hardware equipment such as internal storage and sensors.

Compressed sensing (CS) theory can be used for signal acquisition, encoding and decoding [1]. No matter what type of signals, sparse or compressible representations always exist in the original domain or in some transform domains. During transmission, the linear projection value that far lower than the traditional Nyquist sampling can be used to realize the accurate or high probability reconstruction of the signal. For a discrete signal $x\in R^{n}$, the standard model of CS is
(1)y=Φx,
where $\Phi \in R^{m\times n}\left(m<n\right)$ is a measurement matrix, and $y\in R^{m}$ is the corresponding measurement vector.

It shows that a vector *x* of *n*-dimensional can be compressed into a vector *y* of *m*-dimensional by CS. Therefore, the compression ratio θ can be represented by $\theta =\frac{m}{n}$.

If a measurement vector *y* is given, it is urgently important to reconstruct *x* by measurement matrix Φ. However, this problem is usually NP-hard [2]. If there are less than k(k≪n) non-zero elements in a signal *x*, then the signal *x* is *k*-sparse. Candès and Tao confirmed that if a signal *x* is *k*-sparse and Φ meets the restricted isometry property (RIP), then *y* can accurately reconstruct *x* [3] by solving the following equation,
(2)$$\min_{x\in R^{n}}{\left||x|\right|}_{0}\;\;\;s.t.\;\;y=\Phi x,$$
where ${\left||x|\right|}_{0}=\left|\left\{i|x_{i}\neq 0\right\}\right|$.

Since $l_{1}$-norm is a convex function, it is common method to replace ${\left||x|\right|}_{0}$ with ${\left||x|\right|}_{1}$ in CS, i.e.,
(3)$$\min_{x\in R^{n}}{\left||x|\right|}_{1}\;\;\;s.t.\;\;y=\Phi x,$$
where ${\left||x|\right|}_{1}=|x_{1}\left|+\right|x_{2}\left|+\cdots +\right|x_{n}|$.

For a *k*-sparse signal $x\in R^{n}$, and a matrix $\Phi \in R^{m\times n}$, if there exists a constant $0\leq \delta_{k}<1$ such that
(4)$$\left(1-\delta_{k}\right){\left||x|\right|}_{2}\leq {\left||\Phi x|\right|}_{2}\leq \left(1+\delta_{k}\right){\left||x|\right|}_{2},$$
where ${\left||x|\right|}_{2}^{2}=x_{1}^{2}+x_{2}^{2}+\cdots +x_{n}^{2}$, then the matrix Φ is said to satisfy the RIP of order *k*, and the smallest $\delta_{k}$ is defined as the restricted isometry constant (RIC) of order *k*.

Another important standard is coherence [4] in measurement matrices of CS.

Let $\Phi =(\Phi_{1},\Phi_{2},\cdots ,\Phi_{n})$, where $\Phi_{i}$ is *i*-th column of Φ, 1≤i≤n. Then, the coherence of Φ can be expressed by the following equation
(5)$$\mu \left(\Phi \right)=\max_{i\neq j}\frac{\left|\langle \Phi_{i},\Phi_{j}\rangle \right|}{\left|\right|\Phi_{i}{\left|\right|}_{2}\left|\right|\Phi_{j}{\left|\right|}_{2}},1\leq i,j\leq n,$$
where $\langle \Phi_{i},\Phi_{j}\rangle$ denotes the Hermite inner product of $\Phi_{i}$ and $\Phi_{j}$.

There is a relationship between the coherence and RIP of a matrix as follows.

If Φ is a unit-norm matrix and μ=μ(Φ), then Φ is said to satisfy the RIP of order *k* with $\delta_{k}\leq \mu \left(k-1\right)$ for all $k<\frac{1}{\mu }+1$.

Furthermore, for a matrix Φ with size m×n-dimensional, the coherence of Φ can be represented by Welch bound as follows [5]
(6)$$\mu \left(\Phi \right)\geq \sqrt{\frac{n-m}{m(n-1)}}.$$

The main problem in CS is to find deterministic constructions based on coherence which beat this square root bound.

In CS theory, measurement matrices are not only the vital step to guarantee the quality of signal sampling but also the vital step to determine the difficulty of compressed sampling hardware implementation. There are two main types of measurement matrices. One is random matrices. Random matrices consist of Gaussian matrices, Bernoulli matrices, local Fourier matrices and so on [6,7,8,9,10,11]. Although these matrices can reconstruct the original signals well, they are hard to be implemented in hardware, and the matrix elements require a lot of storage space. Some scholars have proposed using the Toplitz matrices to construct the measurement matrices [12,13]. Although the Toplitz matrices can save some storage space, it is still difficult to be implemented in hardware. Deterministic matrices can improve the transmission efficiency and reduce the storage space [14,15], but they have large reconstruction errors. When constructing this kind of matrices, as long as the system and construction parameters are determined, the size and elements of the matrix will also be determined. DeVore used polynomials over finite field $F_{p}$ to construct measurement matrices in [16]. Li et al. gave a construction method of a sparse measurement matrix based on algebraic curves in [17]. The main tools for constructing deterministic measurement matrices are coding [18,19,20,21,22], geometry over finite fields [23,24,25,26,27,28], design theory [29,30,31,32], and so on.

Compared with CS, for signals of the same size, the advantage of semi-tensor product compressed sensing (STP-CS) is that the number of columns of the measurement matrices can be a factor of CS, which greatly reduces the storage space of measurement matrices. Therefore, we are more interested in the research of STP-CS. The main contribution of the paper is to give a construction of structured random matrices and apply these matrices to STP-CS. The structured random matrices can be obtained by the embedding operation of two seed matrices in which one is determined, and the other is random. In addition, as long as the system and constructed parameters generate structured random matrices, the size of the matrix is determined, but the elements of the matrix are arranged in a structured random manner. When transmitting and storing the matrix, the system, constructed parameters and a random seed matrix need to be transmitted or stored, which can improve the transmission efficiency and reduce the storage scale of a random matrix. Compared with random matrices, the structured random matrices overcome the disadvantage of large storage space of random matrices and is relatively convenient for hardware implementation. Compared with deterministic matrices, the structured random matrices have good reconstruction accuracy. Therefore, a structured random matrix has greater application value in STP-CS model.

Aiming at existing shortcomings—a random matrix needs large storage space and is difficult to be implemented in hardware, and a deterministic matrix has large reconstruction error—the objective of this paper is to find an effective method to balance these performances. The main contributions of our work are summarized as follows:A construction method of structured random matrices is given, where one is the incidence matrices of combinatorial designs, and the other is obtained by the Gram–Schmidt orthonormalization of random matrices.A STP-CS model based on the structured random matrices is proposed.Experimental results indicate that our matrices are more suitable for the reconstruction of one-dimensional signals and two-dimensional images.

The difference between this paper and previous works [14,31] is as follows:The measurement matrices constructed in this paper are structured random matrices, while the measurement matrices constructed in [14,31] are determined matrices.This paper studies STP-CS model, while [14] studies the block compressed sensing model (BCS), and [31] studies CS model.

The details of each section are as follows. Section 2 introduces some related knowledge. Section 3 proposes a new model, which applies the structured random matrices to STP-CS. Section 4 gives simulation experiments, analyzes and compares the performance of our matrices with several famous matrices.

## 2. Preliminaries

In this section, projective geometry [33], balanced incomplete block design [34], embedding operation of binary matrix [35] and semi-tensor product compressed sensing [36] are introduced.

### 2.1. Projective Geometry

Let $F_{q}$ be the finite field with *q* elements. $F_{q}^{\left(n+1\right)}$ is the (n+1)-dimensional row vector space over $F_{q}$, where *q* is a prime power, and *n* is a positive integer. The 1-dimensional, 2-dimensional, 3-dimensional, and *n*-dimensional vector subspaces of $F_{q}^{\left(n+1\right)}$ are called points, lines, planes, and hyperplanes, respectively. In general, the (r+1)-dimensional vector subspaces of $F_{q}^{\left(n+1\right)}$ are called projective *r*-flats, or simply *r*-flats (0≤r≤n). Thus, 0-flats, 1-flats, 2-flats, and (n−1)-flats are points, lines, planes, and hyperplanes, respectively. If an *r*-flat as a vector subspace contains or is contained in an *s*-flat as a vector subspace, then the *r*-flat is called incidented with the *s*-flats. Then, the set of points, i.e., the set of 1-dimensional vector subspaces of $F_{q}^{\left(n+1\right)}$, together with the *r*-flats (0≤r≤n) and the incidence relation among them defined above is said to be the *n*-dimensional projective space over $F_{q}$ and is denoted by $\mathrm{PG}(n,F_{q})$.

### 2.2. Balanced Incomplete Block Design

**Definition** **1.**
*Let v,k,b,r,λ be positive integers, and v≥k≥2. For a finite set $x=\{x_{1},x_{2},\cdots ,x_{v}\}$, a subset family $B=\{B_{1},B_{2},\cdots ,B_{b}\}$ of x, where $x_{1},x_{2},\cdots ,x_{v}$ are called points, $B_{1},B_{2},\cdots ,B_{b}$ are called blocks, if*

*(1) There are k(k<v) points in each block;*

*(2) Each point in x appears in r blocks;*

*(3) Each pair of distinct points is contained in exactly λ blocks.*

*Then (x,B) is called a (v,b,r,k,λ) balanced incomplete block design or simply (v,b,r,k,λ)-BIBD.*


**Definition** **2.**
*For a (v,b,r,k,λ)-BIBD, if b=v (or r=k or $\lambda \left(v-1\right)=k^{2}-k$), then this design is symmetric. Symmetric BIBD is simply denoted by SBIBD.*


### 2.3. Embedding Operation of Binary Matrix

**Definition** **3.***Let $H=(h_{1},h_{2},\cdots ,h_{n})$, where $h_{i}$ is the i-th column of H, $h_{i}$ has d “1", 1≤i≤n. In addition, A is a matrix with size $d\times n_{1}$-dimensional, each element 1 in $h_{i}$ is substitute for a distinct row of A, and each element 0 is substitute for the $1\times n_{1}$ row vector (0,0,⋯,0). The result matrix* Φ *is expressed as*
(7)$$\begin{array}{c} \Phi =H⊙A, \end{array}$$
*and* Φ *is an $m\times n_{1}$-dimensional matrix, where “⊙” denotes the embedding operation of the matrix A in the matrix H.*

The specific process of the above embedding operation is shown in Figure 1.

### 2.4. Semi-Tensor Product Compressed Sensing

**Definition** **4.**
*Let x be a row vector with size np-dimensional and $y={\left[Y_{1},\cdots ,Y_{p}\right]}^{T}$ be a column vector with size p-dimensional. Split x into p blocks, named $x^{1},\cdots ,x^{p}$; the size of each block is n-dimensional. The semi-tensor product (STP) is defined as*

(8)
$$x⋉y=\sum_{i=1}^{p}x^{i}Y_{i}\in R^{1\times n},$$



**Definition** **5.**
*Let $A\in R^{m\times np}$ and $B\in R^{p\times q}$; then, the STP of A and B is defined as follows,*

(9)
C=A⋉B,

*C has m×q blocks as $C=(c_{i,j})$ and each block is*

(10)
$$c_{i,j}=a^{i}⋉b_{j},i=1,2,\cdots ,m,j=1,2,\cdots ,q,$$

*where $a^{i}$ is the i-th row of A and $b_{j}$ is the j-th column of B.*


For a signal $x\in R^{p}$ and a measurement matrix $\Phi \in R^{m\times n}$ (m<n), the STP-CS model [36] is as follows
(11)$$\begin{array}{c} y=\Phi ⋉x, \end{array}$$
where $y\in R^{\frac{mp}{n}}$ and p=lcm(n,p).

Similarly, we can also define the STP-CS by using Kronecker product as follows
(12)$$\begin{array}{c} y=(\Phi ⊗I_{\frac{p}{n}})x, \end{array}$$
where $I_{\frac{p}{n}}$ is a $\frac{p}{n}\times \frac{p}{n}$-dimensional identity matrix, $\frac{p}{n}$ is a positive integer, and “⊗” denotes the Kronecker product.

**Theorem** **1.**
*The measurement matrix $\Phi ⊗I_{\frac{p}{n}}$ has coherence*

(13)
$$\begin{array}{c} \mu \left(\Phi ⊗I_{\frac{p}{n}}\right)=\mu \left(\Phi \right). \end{array}$$



## 3. Construction of Structured Random Measurement Matrices in STP-CS

Compared with CS, for signals of the same size, the advantage of STP-CS is that the number of columns of the measurement matrix can be a factor of CS, which greatly reduces the storage space of measurement matrices. Compared with measurement matrices in STP-CS, the structured random matrices only need to store two seed matrices instead of the whole matrix. To sum up, the structured matrices have lower storage space in STP-CS. In this section, we give a new model that applies the structured random matrices to STP-CS.

### 3.1. Construction of $\left(q^{2}+q+1,q+1,1\right)$-SBIBD

The 1-dimensional projective space over $F_{q}$ only has q+1 points, so it is less interesting. So, let us start our discussion with the 2-dimensional projective planes $\mathrm{PG}(2,F_{q})$. In $\mathrm{PG}(2,F_{q})$, there are $q^{2}+q+1$ points and $q^{2}+q+1$ lines; every line contains q+1 points and every point passes through q+1 lines; any two different points are connected by exactly one line; any two different lines intersect in exactly one point. It is easy to find that

(i) A finite projective plane of order *q* is $\left(q^{2}+q+1,q+1,1\right)$-BIBD. A block is called a line in a finite projective plane.

(ii) For the parameter set $v=q^{2}+q+1$, k=q+1, λ=1 of a BIBD, we must have $r=\frac{\lambda (v-1)}{k-1}=n+1=k$ and, hence, b=v. So, $\left(q^{2}+q+1,q+1,1\right)$-BIBD is necessarily symmetric, and it is simply denoted by $\left(q^{2}+q+1,q+1,1\right)$-SBIBD.

Based on this, for $\left(q^{2}+q+1,q+1,1\right)$-SBIBD, we assume that $x=\{x_{1},x_{2},\cdots ,x_{q^{2}+q+1}\}$ is a set of points, and $B=\{B_{1},B_{2},\cdots ,B_{q^{2}+q+1}\}$ is a set of blocks. The incidence matrix of $\left(q^{2}+q+1,q+1,1\right)$-SBIBD is defined by
(14)$$M={\left(m_{i,j}\right)}_{1\leq i,j\leq q^{2}+q+1},$$
whose rows are marked by $x_{1},x_{2},\cdots ,x_{q^{2}+q+1}$ and columns are marked by $B_{1},B_{2},\cdots ,B_{q^{2}+q+1}$, and
(15)$$m_{i,j}=\left\{\begin{array}{cc} 1, & \;\;\mathrm{if}\;\;x_{i}\in B_{j}\\ 0, & \;\;\mathrm{otherwise} \end{array}\right..$$

Obviously, *M* has the same row-degree and column-degree, both of which are q+1.

**Theorem** **2.**
*If the incidence matrix of $\left(q^{2}+q+1,q+1,1\right)$-SBIBD is M. Then, the matrix M has coherence $\mu \left(M\right)=\frac{1}{q+1}$.*


In the following, the relationship between some known projective planes and BIBD is shown in Table 1.

### 3.2. Gram–Schmidt Orthonormalization

Let $A=(a_{1},a_{2},\cdots ,a_{q+1})$ be a random matrix, where $a_{i}\in R^{\left(q+1\right)}$ denotes the *i*-th column of *A*, 1≤i≤q+1. In order to ensure that the random matrix *A* has small coherence, all columns in matrix *A* are Gram–Schmidt orthonormalization, and the process is as follows

Let $b_{1}=a_{1}$,

$b_{2}=a_{2}-\frac{\langle a_{2},b_{1}\rangle }{\langle b_{1},b_{1}\rangle }b_{1}$,

⋮

$b_{q+1}=a_{q+1}-\sum_{i=1}^{q}\frac{\langle a_{q+1},b_{i}\rangle }{\langle b_{i},b_{i}\rangle }b_{i}$.

Then, $b_{1}$, $b_{2}$, ⋯, $b_{q+1}$ are normalized, i.e.,

$c_{i}=\frac{b_{i}}{|b_{i}|}$,

In this way, we obtain a normalized orthogonal matrix *C* of matrix *A*.

**Remark** **1.**
*According to Definition 3, let $\Phi =M⊙C\in R^{\left(q^{2}+q+1\right)\times \left(q^{3}+2q^{2}+2q+1\right)}$; there are two cases in the following*


*If A is a deterministic matrix, then C must also be deterministic. Therefore,* Φ *is a deterministic matrix;**If A is a random matrix, then C must also be random. Therefore,* Φ *is a structured random matrix.*

There are many researches on deterministic matrices and random matrices, but few on structured random matrices. Combining the advantages of random matrices and the incidence matrices of combinatorial designs, this paper constructs the structured random measurement matrices and applied them in STP-CS.

### 3.3. Sampling Model

In the following, we consider Φ=M⊙C as a measurement matrix in STP-CS. Let *p* be a positive integer and satisfy $p=\mathrm{lcm}(q^{3}+2q^{2}+2q+1,p)$. For a signal $x\in R^{p}$, a novel semi-tensor product compressed sensing model by the embedding operation (STP-CS-EO) is given in the following
(16)$$\begin{array}{ccc} y & = & \Phi ⋉x\\ & = & (\Phi ⊗I_{\frac{p}{q^{3}+2q^{2}+2q+1}})x\\ & = & [\left(M⊙C\right)⊗I_{\frac{p}{q^{3}+2q^{2}+2q+1}}]x, \end{array}$$
then $y\in R^{\frac{p}{q+1}}$.

According to Theorem 1, it finds that
(17)$$\begin{array}{c} \mu \left(\Phi \right)=\mu [\left(M⊙C\right)⊗I_{\frac{p}{q^{3}+2q^{2}+2q+1}}]=\mu \left(M⊙C\right). \end{array}$$

**Remark** **2.**
*Let $x\in R^{N}$ be a discrete signal, where N is a positive integer. For $y\in R^{m}$, we present a comparison of CS, Kronecker product compressed sensing (KP-CS), block compressed sampling based on the embedding operation (BCS-EO), STP-CS, Kronecker product semi-tensor product compressed sensing (KP-STP-CS) and semi-tensor product compressed sensing based on the embedding operation (STP-CS-EO). Table 2 lists the comparison of storage space and sampling complexity of the measurement matrices corresponding to the above six sampling models. Sampling complexity is defined by the multiplication times between a matrix and a vector in the sampling process. For STP-CS, t is a positive integer and satisfies t|m, t|N. For signals of the same size, the advantage of STP-CS is that the number of columns of the measurement matrix can be a factor of CS. For KP-CS and KP-STP-CS, $I_{p}$ is a p×p-dimensional identity matrix, where p is a positive integer and satisfies p|m, p|N. For BCS-EO and STP-CS-EO, $H_{1}$ and $H_{2}$ have column-degree d, and $A_{1}$ and $A_{2}$ have size d×d-dimensional, where d is a positive integer and satisfies d|N. Compared with CS, KP-CS, BCS-EO, STP-CS and KP-STP-CS, the STP-CS-EO model has lower storage space and lower sampling complexity if $t<p,N<\frac{dt^{3}p}{d-p^{2}},m>\frac{d^{3}p^{2}t^{2}}{N(d-p^{2})}$ or if $t<p,N>\frac{dt^{3}p}{d-p^{2}},m>\frac{dp^{2}}{t}$ or $t>p,N>\frac{d^{2}pt^{2}}{\left(d-p^{2}\right)},m>dp$ or $t>p,N<\frac{d^{2}pt^{2}}{\left(d-p^{2}\right)},m>\frac{d^{3}p^{2}t^{2}}{N(d-p^{2})}$.*


In the following, we calculate the coherence of the matrix M⊙C.

**Theorem** **3.**
*Let M be the incidence matrix of $\left(q^{2}+q+1,q+1,1\right)$-SBIBD and $C={\left(c_{s,t}\right)}_{1\leq s,t\leq q+1}$ be a (q+1)×(q+1)-dimensional normalized orthogonal random matrix; then, there is a construction of structured random measurement matrices for a $\left(q^{2}+q+1\right)\times \left(q^{3}+2q^{2}+2q+1\right)$-dimensional matrix Φ=M⊙C with coherence $\mu \left(\Phi \right)=\max|\langle c_{s,t},c_{s_{1},t_{1}}\rangle |$, where $1\leq s,s_{1}\leq q+1,1\leq t,t_{1}\leq q+1$.*


**Proof** **of Theorem 3.**According to Φ=M⊙C, then Φ has size $\left(q^{2}+q+1\right)\times \left(q^{3}+2q^{2}+2q+1\right)$-dimensional. Let $M=(m_{1},m_{2},\cdots ,m_{q+1})$, where $m_{i}$ is the *i*-th column of *M*, i=1,2,⋯,q+1. $C={\left(c_{s,t}\right)}_{1\leq s,t\leq q+1}$ is a (q+1)×(q+1)-dimensional normalized orthogonal random matrix. For any two columns $\Phi_{j_{1}}$ and $\Phi_{j_{2}}$ in Φ,(1) If $\Phi_{j_{1}}$ and $\Phi_{j_{2}}$ correspond to the same column $m_{i_{1}}$ in *M*, then we have
$$\frac{\left|\langle \phi_{j_{1}},\phi_{j_{2}}\rangle \right|}{\left|\right|\phi_{j_{1}}{\left|\right|}_{2}\left|\right|\phi_{j_{2}}{\left|\right|}_{2}}=0,$$
since *C* is a orthogonal matrix;(2) If $\Phi_{j_{1}}$ and $\Phi_{j_{2}}$ correspond to two different columns $m_{i_{1}}$ and $m_{i_{2}}$ in *M*, then we have
$$\frac{\left|\langle \phi_{j_{1}},\phi_{j_{2}}\rangle \right|}{\left|\right|\phi_{j_{1}}{\left|\right|}_{2}\left|\right|\phi_{j_{2}}{\left|\right|}_{2}}=\left|\langle c_{s,t},c_{s_{1},t_{1}}\rangle \right|,$$
since *C* is a normalized matrix, where $c_{s,t}$ and $c_{s_{1},t_{1}}$ are the elements of matrix *C*, $1\leq s,s_{1}\leq q+1,1\leq t,t_{1}\leq q+1$.Therefore, Φ has coherence $\mu \left(\Phi \right)=\max|\langle c_{s,t},c_{s_{1},t_{1}}\rangle |$.

## 4. Experimental Simulation

In this section, our measurement matrices are compared with several famous matrices. Simulation results show that our matrices can be regarded as an effective signal processing method.

### 4.1. Reconstruction of 1-Dimensional Signals

Let *x* be a signal. We select the orthogonal matching pursuit (OMP) [37] algorithm and the basis pursuit (BP) [38] algorithm to solve the $l_{1}$-minimization problem, where the solution is represented by $x^{\prime }$. The definition of the reconstruction Signal-to-Noise Ratio (SNR) of *x* is
(18)$$\begin{array}{c} \mathrm{SNR}\left(x\right)=10\cdot \mathrm{lg}\left(\frac{{\left||x|\right|}_{2}^{2}}{||x-x^{\prime }{\left|\right|}_{2}^{2}}\right)\mathrm{dB}. \end{array}$$

For noiseless recovery, if SNR(x)≥100dB, then the signal *x* is called perfect recovery. For every sparsity order, we reconstruct 1000 noiseless signals to calculate the perfect recovery percentage.

**Example** **1.**
*Let $M_{1}$ be the incidence matrix of (73,9,1)-SBIBD. Then, we construct three structured random measurement matrices $\Phi_{1}=M_{1}⊙C_{1}$, $\Phi_{2}=M_{1}⊙C_{2}$ and $\Phi_{3}=M_{1}⊙C_{3}$, where $C_{1}$, $C_{2}$, $C_{3}$ are a normalized orthogonal matrix of 9×9-dimensional Gaussian, Bernoulli, and Toeplitz matrix, respectively.*

*For measurement matrices $\Phi_{1}⊗I_{2}$, $\Phi_{1}⊗I_{3}$ and $\Phi_{1}⊗I_{4}$, Figure 2a–c show for different sparsity orders the perfect recovery percentages of 1314×1-dimensional, 1971×1-dimensional and 2628×1-dimensional sparse signals, respectively. It shows that the reconstruction effects of $\Phi_{1}⊗I_{2}$, $\Phi_{1}⊗I_{3}$ and $\Phi_{1}⊗I_{4}$ are better than those of $Gaussian\left(73\times 657\right)⊗I_{2}$, $Gaussian\left(73\times 657\right)⊗I_{3}$ and $Gaussian\left(73\times 657\right)⊗I_{4}$ under OMP obviously, respectively, and their reconstruction effects are similar to those of $Gaussian\left(73\times 657\right)⊗I_{2}$, $Gaussian\left(73\times 657\right)⊗I_{3}$ and $Gaussian\left(73\times 657\right)⊗I_{4}$ under BP, respectively.*

*For measurement matrices $\Phi_{2}⊗I_{2}$, $\Phi_{2}⊗I_{3}$ and $\Phi_{2}⊗I_{4}$. Figure 3a–c show for different sparsity orders the perfect recovery percentages of 1314×1-dimensional, 1971×1-dimensional and 2628×1-dimensional sparse signals, respectively. It shows that the reconstruction effects of $\Phi_{2}⊗I_{2}$, $\Phi_{2}⊗I_{3}$ and $\Phi_{2}⊗I_{4}$ are better than those of $Bernoulli\left(73\times 657\right)⊗I_{2}$, $Bernoulli\left(73\times 657\right)⊗I_{3}$ and $Bernoulli\left(73\times 657\right)⊗I_{4}$ under OMP obviously, respectively, and their reconstruction effects are similar to those of $Bernoulli\left(73\times 657\right)⊗I_{2}$, $Bernoulli\left(73\times 657\right)⊗I_{3}$ and $Bernoulli\left(73\times 657\right)⊗I_{4}$ under BP, respectively.*

*For measurement matrices $\Phi_{3}⊗I_{2}$, $\Phi_{3}⊗I_{3}$ and $\Phi_{3}⊗I_{4}$. Figure 4a–c show for different sparsity orders the perfect recovery percentages of 1314×1-dimensional, 1971×1-dimensional and 2628×1-dimensional sparse signals, respectively. It shows that the reconstruction effects of $\Phi_{2}⊗I_{2}$, $\Phi_{2}⊗I_{3}$ and $\Phi_{2}⊗I_{4}$ are better than those of $Toeplitz\left(73\times 657\right)⊗I_{2}$, $Toeplitz\left(73\times 657\right)⊗I_{3}$ and $Toeplitz\left(73\times 657\right)⊗I_{4}$ under OMP obviously, respectively, and their reconstruction effects are similar to those of $Toeplitz\left(73\times 657\right)⊗I_{2}$, $Toeplitz\left(73\times 657\right)⊗I_{3}$ and $Toeplitz\left(73\times 657\right)⊗I_{4}$ under BP, respectively.*


**Example** **2.**
*Let $e\in R^{p}$ be the additive white Gaussian noise with SNR 50 dB. Figure 5 shows the reconstruction SNR comparison of $\Phi_{1}⊗I_{2}$, $\Phi_{1}⊗I_{3}$ and $\Phi_{1}⊗I_{4}$ with Gaussian$\left(73\times 657\right)⊗I_{2}$, Gaussian$\left(73\times 657\right)⊗I_{3}$ and Gaussian$\left(73\times 657\right)⊗I_{4}$ under OMP and BP, respectively. It shows that the reconstruction SNR effects of $\Phi_{1}⊗I_{2}$, $\Phi_{1}⊗I_{3}$ and $\Phi_{1}⊗I_{4}$ are better than those of $Gaussian\left(73\times 657\right)⊗I_{2}$, $Gaussian\left(73\times 657\right)⊗I_{3}$ and $Gaussian\left(73\times 657\right)⊗I_{4}$ under OMP, respectively, and their reconstruction SNR effects are similar to those of $Gaussian\left(73\times 657\right)⊗I_{2}$, $Gaussian\left(73\times 657\right)⊗I_{3}$ and $Gaussian\left(73\times 657\right)⊗I_{4}$ under BP, respectively.*

*Figure 6 shows the reconstruction SNR comparison of $\Phi_{2}⊗I_{2}$, $\Phi_{2}⊗I_{3}$ and $\Phi_{2}⊗I_{4}$ with $Bernoulli\left(73\times 657\right)⊗I_{2}$, $Bernoulli\left(73\times 657\right)⊗I_{3}$ and $Bernoulli\left(73\times 657\right)⊗I_{4}$ under OMP and BP, respectively. It shows that the reconstruction SNR effects of $\Phi_{2}⊗I_{2}$, $\Phi_{2}⊗I_{3}$ and $\Phi_{2}⊗I_{4}$ are better than those of $Bernoulli\left(73\times 657\right)⊗I_{2}$, $Bernoulli\left(73\times 657\right)⊗I_{3}$ and $Bernoulli\left(73\times 657\right)⊗I_{4}$ under OMP, respectively, and their reconstruction SNR effects are similar to those of $Bernoulli\left(73\times 657\right)⊗I_{2}$, $Bernoulli\left(73\times 657\right)⊗I_{3}$ and $Bernoulli\left(73\times 657\right)⊗I_{4}$ under BP, respectively.*

*Figure 7 shows the reconstruction SNR comparison of $\Phi_{3}⊗I_{2}$, $\Phi_{3}⊗I_{3}$ and $\Phi_{3}⊗I_{4}$ with Toeplitz$\left(73\times 657\right)⊗I_{2}$, Toeplitz$\left(73\times 657\right)⊗I_{3}$ and Toeplitz$\left(73\times 657\right)⊗I_{4}$ under OMP and BP, respectively. It shows that the reconstruction SNR effects of $\Phi_{2}⊗I_{2}$, $\Phi_{2}⊗I_{3}$ and $\Phi_{2}⊗I_{4}$ are better than those of $Toeplitz\left(73\times 657\right)⊗I_{2}$, $Toeplitz\left(73\times 657\right)⊗I_{3}$ and $Toeplitz\left(73\times 657\right)⊗I_{4}$ under OMP, respectively, and their reconstruction SNR effects are similar to those of $Toeplitz\left(73\times 657\right)⊗I_{2}$, $Toeplitz\left(73\times 657\right)⊗I_{3}$ and $Toeplitz\left(73\times 657\right)⊗I_{4}$ under BP, respectively.*


In applications, the original signal is always disturbed by channel noise. For noisy recovery, the original signal $x\in R^{p}$ is polluted by additive white Gaussian noise $e\in R^{p}$. Therefore, if $\Phi \in R^{\left(q^{2}+q+1\right)\times \left(q^{3}+2q^{2}+2q+1\right)}$ is a measurement matrix, then
(19)$$\begin{array}{ccc} y & = & \Phi ⋉(x+e)\\ & = & \left[\Phi ⊗I_{\frac{p}{q^{3}+2q^{2}+2q+1}}\right]\left(x+e\right), \end{array}$$
where $y\in R^{\frac{p}{q+1}}$ and $p=\mathrm{lcm}(q^{3}+2q^{2}+2q+1,p)$. For every sparsity order, we calculate the reconstruction SNR by reconstructing 1000 noisy signals.

Furthermore, the original signals usually approach to sparse, and the measurement vector may also be polluted by the noise in the measurement domain. Hence, we study the noise recovery effect of our matrices in the actual STP-CS,
(20)$$\begin{array}{c} y=\Phi ⋉\left(x+e_{d}\right)+e_{m}, \end{array}$$
where $e_{m}\in R^{\frac{p}{q+1}}$ denotes noise in the measurement domain, and $e_{d}\in R^{p}$ denotes noise in the data-domain.

**Example** **3.**
*Let $e_{d}\in R^{p}$, $e_{m}\in R^{\frac{p}{q+1}}$ be the additive white Gaussian noise with SNR 20–100 dB. Figure 8, Figure 9 and Figure 10 show the comparison average recovery SNR for $\Phi_{1}⊗I_{2}$, $\Phi_{2}⊗I_{2}$ and $\Phi_{3}⊗I_{2}$ with Gaussian$\left(73\times 657\right)⊗I_{2}$, Bernoulli$\left(73\times 657\right)⊗I_{2}$ and Toeplitz$\left(73\times 657\right)⊗I_{2}$ under OMP and BP, respectively. The stable and robust empirical effects of $\Phi_{1}⊗I_{2}$, $\Phi_{2}⊗I_{2}$ and $\Phi_{3}⊗I_{2}$ are similar to Gaussian$\left(73\times 657\right)⊗I_{2}$, Bernoulli$\left(73\times 657\right)⊗I_{2}$ and Toeplitz$\left(73\times 657\right)⊗I_{2}$, respectively.*


### 4.2. Reconstruction of 2-Dimensional Images

In this subsection, we select the orthogonal matching pursuit (OMP) algorithm, basis pursuit (BP) algorithm, iterative soft thresholding (IST) [39] algorithm and subspace pursuit (SP) [40] algorithm for testing. When CS reconstructs a gray image, it is hard to judge the distortion of the reconstructed image by the naked eye and other subjective ways. Hence, it is necessary to give an important parameter to truly evaluate the quality of the reconstructed image; that is, the definition of peak signal-to-noise ratio (PSNR) is as follows:(21)$$\mathrm{PSNR}=10\cdot \mathrm{lg}(\frac{255^{2}}{\mathrm{MSE}})\mathrm{dB},$$
where MSE represents the normalized mean square error, that is
(22)$$\mathrm{MSE}=\frac{1}{M\times N}\sum \sum {\left[\Psi \left(x,y\right)-\Psi^{\prime }\left(x,y\right)\right]}^{2},$$
where M×N represents the image size, and $\Psi \left(x,y\right),\Psi^{\prime }\left(x,y\right)$ are the gray values of the original image and the reconstructed image at the point (x,y), respectively.

**Example** **4.**
*Let $M_{2}$ be the incidence matrix of (21,5,1)-SBIBD; we construct three structured random measurement matrices $\Phi_{4}=M_{2}⊙C_{4}$, $\Phi_{5}=M_{2}⊙C_{5}$ and $\Phi_{6}=M_{2}⊙C_{6}$, where $C_{4}$, $C_{5}$ and $C_{6}$ are the normalized orthogonal matrix of a 5×5-dimensional Gaussian matrix, Bernoulli matrix, Toeplitz matrix, respectively. Therefore, $\Phi_{4}$, $\Phi_{5}$ and $\Phi_{6}$ are 21×105-dimensional matrices. We consider the matrices $\Phi_{4}⊗I_{2}$, $\Phi_{5}⊗I_{2}$ and $\Phi_{6}⊗I_{2}$ are used to reconstruct four images with size 210×210-dimensional, $\Phi_{4}⊗I_{3}$, $\Phi_{5}⊗I_{3}$ and $\Phi_{6}⊗I_{3}$ are used to reconstruct four images with size 315×315-dimensional, $\Phi_{4}⊗I_{4}$, $\Phi_{5}⊗I_{4}$ and $\Phi_{6}⊗I_{4}$ are used to reconstruct four images with size 420×420-dimensional in Figure 11. Table 3, Table 4 and Table 5 have listed the PSNRs and CPU time of four images in the reconstruction process. It shows that the PSNRs of our measurement matrices are not less than that of the Gaussian matrix, Bernoulli matrix and Toeplitz matrix, under OMP, BP, IST, and SP, respectively. The CPU times of our measurement matrices are not longer than those of the Gaussian matrix, Bernoulli matrix and Toeplitz matrix, under OMP, BP, IST, and SP, respectively.*


## 5. Conclusions

The construction of measurement matrices is not only the vital step to guarantee the quality of signal sampling but also the vital step to determine the difficulty of compressed sampling hardware implementation. Aiming at the present shortcomings—that a random matrix needs large storage space and is difficult to be implemented in hardware, and a deterministic measurement matrix has large reconstruction error—this paper constructs a structured random matrix by the embedding operation of two seed matrices in which one is the incidence matrix of $\left(q^{2}+q+1,q+1,1\right)$-SBIBD, and the other is obtained by Gram–Schmidt orthonormalization of a (q+1)×(q+1)-dimensional random matrix. Meanwhile, we provide a new model that applies the structured random matrices to semi-tensor product compressed sensing. Finally, compared with the reconstruction effect of several famous matrices, our matrices are more suitable for the reconstruction of one-dimensional signals and two-dimensional images by experimental simulation. In addition, due to randomness, low storage space and shorter reconstruction time, our matrices have good performances in the reconstruction of signals and images. To sum up, the perspectives to improve the performance of the method are as follows:(1)Special structure of the incidence matrix of $\left(q^{2}+q+1,q+1,1\right)$-SBIBD;(2)Gram–Schmidt orthonormalization of (q+1)×(q+1)-dimensional random matrix,(3)Semi-tensor product compressed sensing based on the structured random matrices.

## Figures and Tables

**Figure 1 sensors-22-08260-f001:**
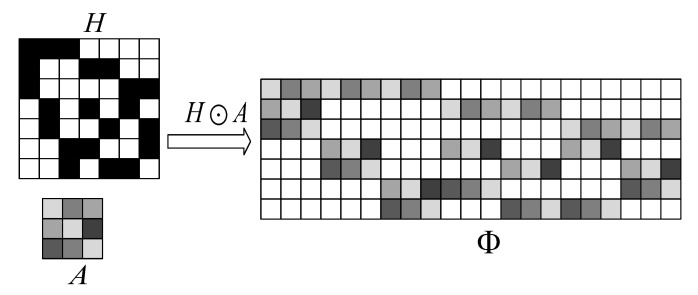
The specific process of *A* as the embedding matrix in matrix *H*.

**Figure 2 sensors-22-08260-f002:**
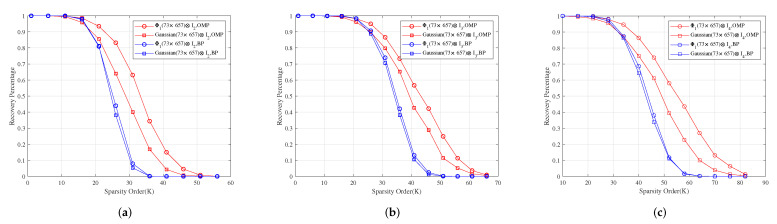
The relationship between the perfect recovery percentage and sparsity order of sparse signals under OMP and BP. $\Phi_{1}⊗I_{2}$, $\Phi_{1}⊗I_{3}$ and $\Phi_{1}⊗I_{4}$ are the corresponding measurement matrices in (**a**–**c**), respectively.

**Figure 3 sensors-22-08260-f003:**
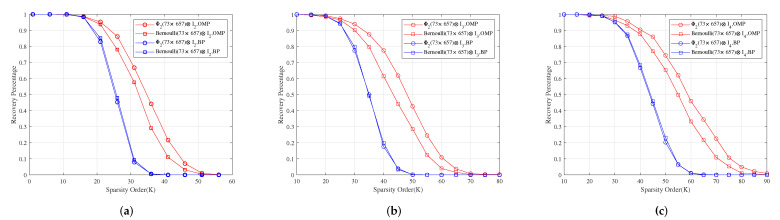
The relationship between the perfect recovery percentage and sparsity order of sparse signals under OMP and BP. $\Phi_{2}⊗I_{2}$, $\Phi_{2}⊗I_{3}$ and $\Phi_{2}⊗I_{4}$ are the corresponding measurement matrices in (**a**–**c**), respectively.

**Figure 4 sensors-22-08260-f004:**
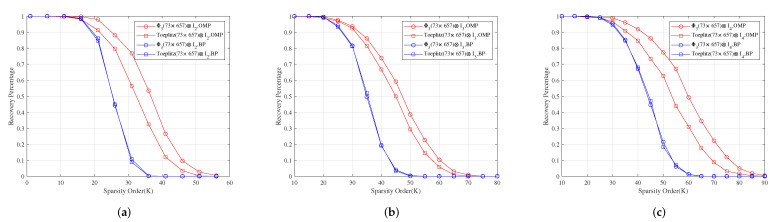
The relationship between the perfect recovery percentage and sparsity order of sparse signals under OMP and BP. $\Phi_{3}⊗I_{2}$, $\Phi_{3}⊗I_{3}$ and $\Phi_{3}⊗I_{4}$ are the corresponding measurement matrices in (**a**–**c**), respectively.

**Figure 5 sensors-22-08260-f005:**
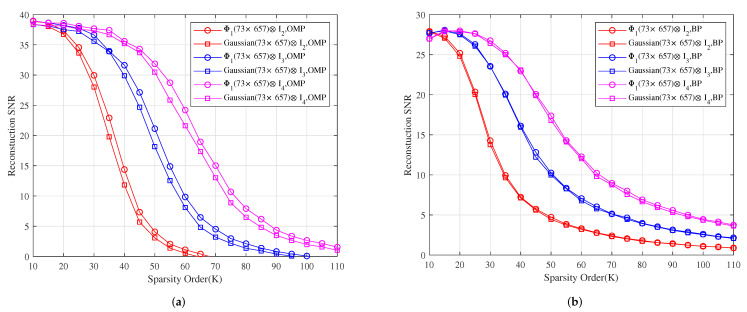
The relationship between the reconstruction SNR and sparsity order of sparse signals under OMP and BP. (**a**) The reconstruction SNR comparison of $\Phi_{1}⊗I_{2}$, $\Phi_{1}⊗I_{3}$ and $\Phi_{1}⊗I_{4}$ with Gaussian$\left(73\times 657\right)⊗I_{2}$, Gaussian$\left(73\times 657\right)⊗I_{3}$ and Gaussian$\left(73\times 657\right)⊗I_{4}$ under OMP, respectively. (**b**) The reconstruction SNR comparison of $\Phi_{1}⊗I_{2}$, $\Phi_{1}⊗I_{3}$ and $\Phi_{1}⊗I_{4}$ with Gaussian$\left(73\times 657\right)⊗I_{2}$, Gaussian$\left(73\times 657\right)⊗I_{3}$ and Gaussian$\left(73\times 657\right)⊗I_{4}$ under BP, respectively.

**Figure 6 sensors-22-08260-f006:**
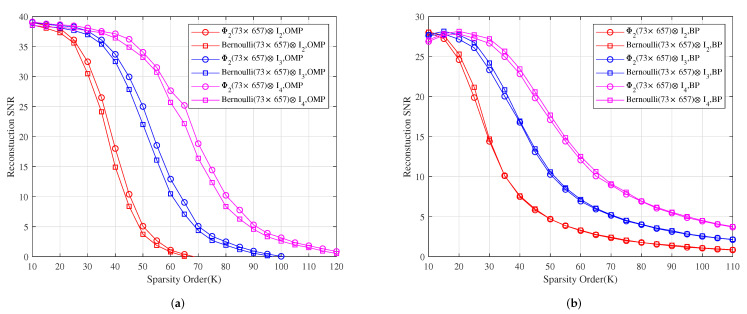
The relationship between the reconstruction SNR and sparsity order of sparse signals under OMP and BP. (**a**) The reconstruction SNR comparison of $\Phi_{2}⊗I_{2}$, $\Phi_{2}⊗I_{3}$ and $\Phi_{2}⊗I_{4}$ with Bernoulli$\left(73\times 657\right)⊗I_{2}$, Bernoulli$\left(73\times 657\right)⊗I_{3}$ and Bernoulli$\left(73\times 657\right)⊗I_{4}$ under OMP, respectively. (**b**) The reconstruction SNR comparison of $\Phi_{2}⊗I_{2}$, $\Phi_{2}⊗I_{3}$ and $\Phi_{2}⊗I_{4}$ with Bernoulli$\left(73\times 657\right)⊗I_{2}$, Bernoulli$\left(73\times 657\right)⊗I_{3}$ and Bernoulli$\left(73\times 657\right)⊗I_{4}$ under BP, respectively.

**Figure 7 sensors-22-08260-f007:**
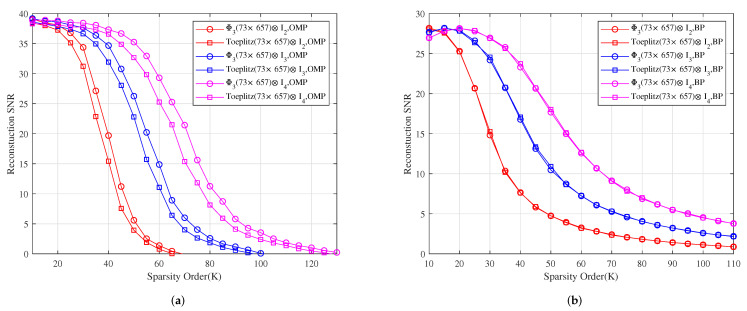
The relationship between the reconstruction SNR and sparsity order of sparse signals under OMP and BP. (**a**) The reconstruction SNR comparison of $\Phi_{3}⊗I_{2}$, $\Phi_{3}⊗I_{3}$ and $\Phi_{3}⊗I_{4}$ with Toeplitz$\left(73\times 657\right)⊗I_{2}$, Toeplitz$\left(73\times 657\right)⊗I_{3}$ and Toeplitz$\left(73\times 657\right)⊗I_{4}$ under OMP, respectively. (**b**) The reconstruction SNR comparison of $\Phi_{3}⊗I_{2}$, $\Phi_{3}⊗I_{3}$ and $\Phi_{3}⊗I_{4}$ with Toeplitz$\left(73\times 657\right)⊗I_{2}$, Toeplitz$\left(73\times 657\right)⊗I_{3}$ and Toeplitz$\left(73\times 657\right)⊗I_{4}$ under BP, respectively.

**Figure 8 sensors-22-08260-f008:**
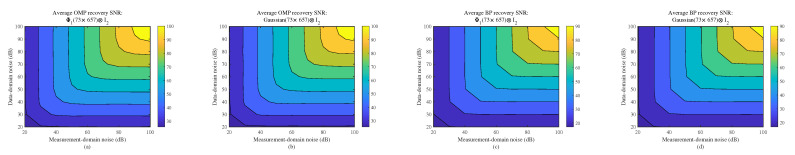
For sparsity order k=9, the relationship of average recovery SNR and noise in measurement domain and data domain. (**a**) Average recovery SNR of $\Phi_{1}⊗I_{2}$ as the measurement matrix under OMP. (**b**) Average recovery SNR of Gaussian$\left(73\times 657\right)⊗I_{2}$ as the measurement matrix under OMP. (**c**) Average recovery SNR of $\Phi_{1}⊗I_{2}$ as the measurement matrix under BP. (**d**) Average recovery SNR of Gaussian$\left(73\times 657\right)⊗I_{2}$ as the measurement matrix under BP.

**Figure 9 sensors-22-08260-f009:**
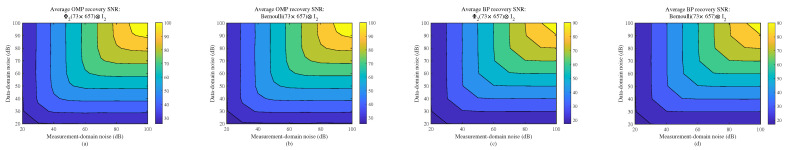
For sparsity order k=9, the relationship of average recovery SNR and noise in measurement domain and data domain. (**a**) Average recovery SNR of $\Phi_{2}⊗I_{2}$ as the measurement matrix under OMP. (**b**) Average recovery SNR of Bernoulli$\left(73\times 657\right)⊗I_{2}$ as the measurement matrix under OMP. (**c**) Average recovery SNR of $\Phi_{2}⊗I_{2}$ as the measurement matrix under BP. (**d**) Average recovery SNR of Bernoulli$\left(73\times 657\right)⊗I_{2}$ as the measurement matrix under BP.

**Figure 10 sensors-22-08260-f010:**
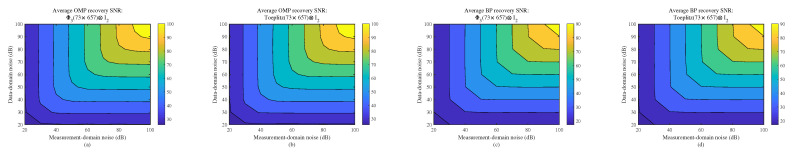
For sparsity order k=9, the relationship of average recovery SNR and noise in measurement domain and data domain. (**a**) Average recovery SNR of $\Phi_{3}⊗I_{2}$ as the measurement matrix under OMP. (**b**) Average recovery SNR of Toeplitz$\left(73\times 657\right)⊗I_{2}$ as the measurement matrix under OMP. (**c**) Average recovery SNR of $\Phi_{3}⊗I_{2}$ as the measurement matrix under BP. (**d**) Average recovery SNR of Toeplitz$\left(73\times 657\right)⊗I_{2}$ as the measurement matrix under BP.

**Figure 11 sensors-22-08260-f011:**
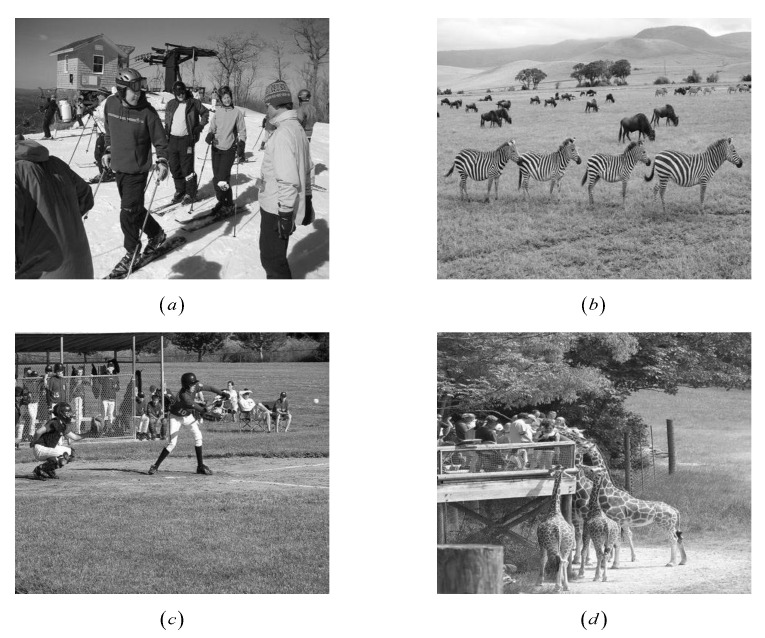
Four test images randomly selected from MSCOCO dataset. (**a**) COCO_val2014_ 000000000761. (**b**) COCO_val2014_000000004754. (**c**) COCO_val2014_000000008119. (**d**) COCO_ val2014_000000193121.

**Table 1 sensors-22-08260-t001:** The relationship between some known projective planes and BIBD.

No.	Order of Projective Planes	Parameters of a BIBD					Coherence
		*v*	*b*	*r*	*k*	*λ*	
1	2	7	7	3	3	1	1/3
2	3	13	13	4	4	1	1/4
3	4	21	21	5	5	1	1/5
4	5	31	31	6	6	1	1/6
5	7	57	57	8	8	1	1/8
6	8	73	73	9	9	1	1/9
7	9	91	91	10	10	1	1/10
8	11	133	133	12	12	1	1/12
9	13	183	183	14	14	1	1/14
10	16	273	273	17	17	1	1/17
11	17	307	307	18	18	1	1/18
12	19	381	381	20	20	1	1/20
13	23	553	553	24	24	1	1/24
14	25	651	651	26	26	1	1/26
15	29	871	871	30	30	1	1/30

**Table 2 sensors-22-08260-t002:** The comparison of storage space and sampling complexity of the measurement matrices corresponding to six sampling models.

Type	Sampling Model	Storage Matrix	Sampling Complexity	Storage Space
CS	$y=\Phi_{1}x$	$\Phi_{1}\in R^{m\times N}$	mN	mN
KP-CS	$y=(P_{1}⊗I_{p})x$	$P_{1}\in R^{\frac{m}{p}\times \frac{N}{p}}$	$\frac{mN}{p^{2}}$	$\frac{mN}{p}$
BCS-EO	$y=(H_{1}⊙A_{1})x$	$H_{1}\in R^{m\times \frac{N}{d}}$ $A_{1}\in R^{d\times d}$	dN	$\frac{mN}{d}+d^{2}$
STP-CS	$y=\Phi_{2}⋉x$	$\Phi_{2}\in R^{\frac{m}{t}\times \frac{N}{t}}$	$\frac{mN}{t}$	$\frac{mN}{t^{2}}$
KP-STP-CS	$y=(P_{2}⊗I_{p})⋉x$	$P_{2}\in R^{\frac{m}{pt}\times \frac{N}{pt}}$	$\frac{mN}{pt}$	$\frac{mN}{p^{2}t^{2}}$
STP-CS-EO	$y=(H_{2}⊙A_{2})⋉x$	$H_{2}\in R^{\frac{m}{t}\times \frac{N}{td}}A_{2}\in R^{d\times d}$	$\frac{dN}{t}$	$\frac{mN}{dt^{2}}+d^{2}$

**Table 3 sensors-22-08260-t003:** The PSNRs of four images and the CPU time of the measurement matrices $\Phi_{4}\left(21\times 105\right)⊗I_{2}$, $\Phi_{5}\left(21\times 105\right)⊗I_{2}$ and $\Phi_{6}\left(21\times 105\right)⊗I_{2}$ in the process of reconstruction.

Algorithm	Measurement Matrix	Image (*a*)	Image (*b*)	Image (*c*)	Image (*d*)
OMP	$\Phi_{4}\left(21\times 105\right)⊗I_{2}$	28.00|0.07	28.16|0.07	28.26|0.07	28.14|0.08
	Gaussian$\left(21\times 105\right)⊗I_{2}$	27.65|0.08	27.96|0.08	27.87|0.09	27.87|0.08
	$\Phi_{5}\left(21\times 105\right)⊗I_{2}$	28.19|0.07	28.34|0.07	28.59|0.07	28.46|0.07
	Bernoulli$\left(21\times 105\right)⊗I_{2}$	28.15|0.08	28.33|0.08	28.55|0.08	28.16|0.08
	$\Phi_{6}\left(21\times 105\right)⊗I_{2}$	28.06|0.07	28.13|0.07	28.39|0.07	27.92|0.07
	Toeplitz$\left(21\times 105\right)⊗I_{2}$	27.87|0.08	27.93|0.07	27.99|0.08	27.78|0.07
BP	$\Phi_{4}\left(21\times 105\right)⊗I_{2}$	27.89|1.54	27.73|1.55	28.00|1.56	27.90|1.54
	Gaussian$\left(21\times 105\right)⊗I_{2}$	27.30|1.89	27.53|1.91	27.65|1.89	27.71|1.91
	$\Phi_{5}\left(21\times 105\right)⊗I_{2}$	28.17|1.54	28.42|1.55	28.47|1.55	28.15|1.93
	Bernoulli$\left(21\times 105\right)⊗I_{2}$	28.15|1.93	28.24|1.93	28.36|1.94	28.09|1.97
	$\Phi_{6}\left(21\times 105\right)⊗I_{2}$	28.03|1.52	27.90|1.51	28.02|1.56	28.03|1.57
	Toeplitz$\left(21\times 105\right)⊗I_{2}$	27.94|2.02	27.60|1.99	27.76|2.07	27.67|2.03
IST	$\Phi_{4}\left(21\times 105\right)⊗I_{2}$	28.00|0.42	28.03|0.40	28.30|0.41	28.22|0.42
	Gaussian$\left(21\times 105\right)⊗I_{2}$	27.86|0.45	28.00|0.44	27.92|0.43	28.03|0.44
	$\Phi_{5}\left(21\times 105\right)⊗I_{2}$	28.12|0.39	28.17|0.39	28.82|0.41	28.28|0.39
	Bernoulli$\left(21\times 105\right)⊗I_{2}$	28.08|0.39	28.03|0.39	27.82|0.45	28.26|0.40
	$\Phi_{6}\left(21\times 105\right)⊗I_{2}$	27.98|0.41	28.21|0.40	28.10|0.41	28.16|0.41
	Toeplitz$\left(21\times 105\right)⊗I_{2}$	27.78|0.44	27.57|0.40	28.09|0.43	28.12|0.41
SP	$\Phi_{4}\left(21\times 105\right)⊗I_{2}$	27.96|0.20	28.80|0.21	28.29|0.21	28.19|0.21
	Gaussian$\left(21\times 105\right)⊗I_{2}$	27.88|0.22	27.37|0.23	28.07|0.22	27.65|0.22
	$\Phi_{5}\left(21\times 105\right)⊗I_{2}$	27.87|0.23	28.14|0.22	27.92|0.22	28.23|0.23
	Bernoulli$\left(21\times 105\right)⊗I_{2}$	27.74|0.23	28.13|0.22	27.88|0.22	28.02|0.23
	$\Phi_{6}\left(21\times 105\right)⊗I_{2}$	28.05|0.21	28.07|0.20	28.06|0.22	28.09|0.21
	Toeplitz$\left(21\times 105\right)⊗I_{2}$	27.69|0.21	27.29|0.22	27.83|0.22	27.78|0.21

**Table 4 sensors-22-08260-t004:** The PSNRs of four images and the CPU time of the measurement matrices $\Phi_{4}\left(21\times 105\right)⊗I_{3}$, $\Phi_{5}\left(21\times 105\right)⊗I_{3}$ and $\Phi_{6}\left(21\times 105\right)⊗I_{3}$ in the process of reconstruction.

Algorithm	Measurement Matrix	Image (*a*)	Image (*b*)	Image (*c*)	Image (*d*)
OMP	$\Phi_{4}\left(21\times 105\right)⊗I_{3}$	27.92|0.11	28.17|0.12	28.41|0.12	28.15|0.12
	Gaussian$\left(21\times 105\right)⊗I_{3}$	27.86|0.12	28.13|0.12	27.96|0.12	28.07|0.13
	$\Phi_{5}\left(21\times 105\right)⊗I_{3}$	27.96|0.12	28.11|0.12	28.27|0.12	28.11|0.11
	Bernoulli$\left(21\times 105\right)⊗I_{3}$	27.90|0.12	27.43|0.13	28.23|0.13	28.04|0.12
	$\Phi_{6}\left(21\times 105\right)⊗I_{3}$	28.07|0.12	28.17|0.12	28.29|0.12	28.36|0.12
	Toeplitz$\left(21\times 105\right)⊗I_{3}$	28.01|0.13	28.12|0.12	27.94|0.12	28.18|0.12
BP	$\Phi_{4}\left(21\times 105\right)⊗I_{3}$	28.07|2.66	28.19|2.65	28.25|2.62	28.14|2.68
	Gaussian$\left(21\times 105\right)⊗I_{3}$	28.05|3.59	28.09|3.37	27.66|3.37	27.92|3.36
	$\Phi_{5}\left(21\times 105\right)⊗I_{3}$	28.29|2.66	28.29|2.99	28.19|2.57	28.28|3.26
	Bernoulli$\left(21\times 105\right)⊗I_{3}$	27.88|3.77	28.05|3.70	27.64|3.74	28.22|3.73
	$\Phi_{6}\left(21\times 105\right)⊗I_{3}$	28.20|2.77	28.28|3.04	28.24|2.56	27.96|2.63
	Toeplitz$\left(21\times 105\right)⊗I_{3}$	28.09|3.81	27.62|3.63	28.16|3.63	27.85|3.57
IST	$\Phi_{4}\left(21\times 105\right)⊗I_{3}$	28.10|0.89	28.29|0.89	28.24|0.91	28.33|0.89
	Gaussian$\left(21\times 105\right)⊗I_{3}$	27.93|0.92	28.19|0.90	27.98|1.00	28.28|0.95
	$\Phi_{5}\left(21\times 105\right)⊗I_{3}$	28.00|0.88	28.34|0.84	28.21|0.85	28.10|0.83
	Bernoulli$\left(21\times 105\right)⊗I_{3}$	27.85|0.88	28.22|0.88	27.95|0.85	28.02|0.85
	$\Phi_{6}\left(21\times 105\right)⊗I_{3}$	28.06|0.88	28.12|0.88	28.33|0.90	28.17|0.89
	Toeplitz$\left(21\times 105\right)⊗I_{3}$	27.91|0.89	27.92|0.93	27.90|0.92	28.00|0.98
SP	$\Phi_{4}\left(21\times 105\right)⊗I_{3}$	28.05|0.33	28.29|0.33	28.47|0.33	28.01|0.33
	Gaussian$\left(21\times 105\right)⊗I_{3}$	27.86|0.38	27.84|0.36	28.25|0.35	27.13|0.35
	$\Phi_{5}\left(21\times 105\right)⊗I_{3}$	27.83|0.34	28.16|0.35	27.81|0.33	27.93|0.34
	Bernoulli$\left(21\times 105\right)⊗I_{3}$	27.75|0.42	28.05|0.35	27.72|0.34	27.87|0.37
	$\Phi_{6}\left(21\times 105\right)⊗I_{3}$	27.99|0.34	28.19|0.34	27.92|0.33	27.96|0.34
	Toeplitz$\left(21\times 105\right)⊗I_{3}$	27.98|0.35	27.94|0.34	27.80|0.34	27.70|0.35

**Table 5 sensors-22-08260-t005:** The PSNRs of four images and the CPU time of the measurement matrices $\Phi_{4}\left(21\times 105\right)⊗I_{4}$, $\Phi_{5}\left(21\times 105\right)⊗I_{4}$ and $\Phi_{6}\left(21\times 105\right)⊗I_{4}$ in the process of reconstruction.

Algorithm	Measurement Matrix	Image (*a*)	Image (*b*)	Image (*c*)	Image (*d*)
OMP	$\Phi_{4}\left(21\times 105\right)⊗I_{4}$	28.16|0.18	28.20|0.19	28.24|0.18	28.18|0.19
	Gaussian$\left(21\times 105\right)⊗I_{4}$	27.94|0.19	28.13|0.19	28.03|0.18	28.08|0.19
	$\Phi_{5}\left(21\times 105\right)⊗I_{4}$	28.17|0.17	28.20|0.17	28.38|0.18	28.10|0.16
	Bernoulli$\left(21\times 105\right)⊗I_{4}$	27.88|0.19	28.02|0.18	28.16|0.19	28.01|0.18
	$\Phi_{6}\left(21\times 105\right)⊗I_{4}$	28.13|0.18	28.14|0.18	28.27|0.18	28.25|0.18
	Toeplitz$\left(21\times 105\right)⊗I_{4}$	27.86|0.20	28.11|0.18	28.13|0.19	28.19|0.19
BP	$\Phi_{4}\left(21\times 105\right)⊗I_{4}$	28.13|3.83	28.21|3.78	28.28|3.81	28.18|3.81
	Gaussian$\left(21\times 105\right)⊗I_{4}$	28.07|5.10	28.10|5.04	27.83|5.06	28.13|5.05
	$\Phi_{5}\left(21\times 105\right)⊗I_{4}$	28.18|4.33	28.17|4.26	27.97|4.23	28.10|3.81
	Bernoulli$\left(21\times 105\right)⊗I_{4}$	28.05|4.34	28.06|4.31	27.71|4.25	27.75|4.33
	$\Phi_{6}\left(21\times 105\right)⊗I_{4}$	28.04|3.72	28.21|3.59	28.21|3.70	28.22|3.82
	Toeplitz$\left(21\times 105\right)⊗I_{4}$	27.95|4.31	27.94|4.30	28.17|4.35	28.14|4.26
IST	$\Phi_{4}\left(21\times 105\right)⊗I_{4}$	28.02|1.56	28.27|1.56	28.17|1.60	28.06|1.81
	Gaussian$\left(21\times 105\right)⊗I_{4}$	27.80|1.71	28.13|1.62	27.96|1.62	27.90|1.53
	$\Phi_{5}\left(21\times 105\right)⊗I_{4}$	28.15|1.63	28.09|1.58	28.01|1.58	28.12|1.63
	Bernoulli$\left(21\times 105\right)⊗I_{4}$	28.04|1.63	27.89|1.62	27.93|1.60	27.85|1.64
	$\Phi_{6}\left(21\times 105\right)⊗I_{4}$	28.10|1.58	28.23|1.61	28.18|1.55	28.09|1.58
	Toeplitz$\left(21\times 105\right)⊗I_{4}$	27.87|1.62	28.12|1.64	28.00|1.65	28.03|1.66
SP	$\Phi_{4}\left(21\times 105\right)⊗I_{4}$	28.26|0.48	28.71|0.48	28.29|0.48	28.20|0.48
	Gaussian$\left(21\times 105\right)⊗I_{4}$	27.96|0.51	28.04|0.51	28.14|0.50	27.98|0.50
	$\Phi_{5}\left(21\times 105\right)⊗I_{4}$	28.10|0.52	28.04|0.49	28.24|0.50	28.24|0.52
	Bernoulli$\left(21\times 105\right)⊗I_{4}$	28.08|0.52	27.90|0.49	28.05|0.56	27.91|0.52
	$\Phi_{6}\left(21\times 105\right)⊗I_{4}$	28.14|0.49	28.27|0.49	28.24|0.49	28.13|0.50
	Toeplitz$\left(21\times 105\right)⊗I_{4}$	27.85|0.50	28.08|0.50	28.08|0.50	28.11|0.51

## Data Availability

Not applicable.

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
