# Peer review of "Construction of Structured Random Measurement Matrices in Semi-Tensor Product Compressed Sensing Based on Combinatorial Designs"

_sensors, 2022, doi:10.3390/s22218260_

Round 1
Reviewer 1 Report
This paper proposes a construction method of structured random measurement matrices in semi-tensor product compressed sensing by combining the advantages of the incidence matrix of combinatorial designs and random matrix. In practice, the random matrix requires a large storage space and its hardware implementation is difficult to implement, while the deterministic matrix has a large reconstruction error. In this work, the author constructs a structured random matrix by the integration operation of two original matrices, one of which is the incidence matrix of the combinatorial planes and the other is the one obtained by Gram-Schmidt orthonormalization of the random matrix. By experimental simulation, the author showed that the constructed matrices are more suitable for the reconstruction of one-dimensional signals and two-dimensional images. This paper well written and the results of the experiments are varied and well presented; nevertheless, some critical issues need to addressed:
- The abstract of the article should be rewritten with emphasis on the objective and contributions of the study.
- The author should clarify the contribution of this study and the difference with previous studies developed in the works of references [14] and [31].
- In Figure 1, please explains the process of integration of the matrix A in the matrix H.
- Justify the choice of the 5x5 size used in the matrices of the simulations of example 4 compared to the 512x512 size of the test images. Can we increase the variation?
- In the conclusion, please give perspectives to improve the performance of the method.

Author Response
Dear Reviewer:
Thank you for your comments on our paper. We appreciate your valuable suggestions. According to your comments, we have revised our paper carefully. And we have highlighted the changes so that you could be easily tracked in the revised manuscript.
Sincerely yours,
All the co-authors

Reviewer 2 Report
Reconstruction of incomplete signals from compressed sensing had been well studied, and there are lots of plenty of previous works in the literature. I think the authors should do the following things to prove the usefully of there work:
1) Use real datasets to do the reconstruction instead of just simulated data and old images;
2) Using and compare with other related the reconstruction algorithms instead of just OMP and BP.
Author Response
Dear Reviewer:
Thank you for your comments on our paper. We really appreciate what you have done. We have carefully studied your valuable suggestions. And we have highlighted the changes so that you could be easily tracked in the revised manuscript.
Sincerely yours,
All the co-authors

Round 2
Reviewer 2 Report
The authors addressed my concerns, and I am happy to accept it.
Author Response
Dear Editor:
Thank you for your comments on our paper. We appreciate your valuable suggestions because they are very helpful in guiding our research. According to your suggestions, we have revised our paper carefully. The comments and the corresponding modifications are listed in cover letter.
Sincerely yours,
All the co-authors
